# T-2 Toxin Induces Oxidative Stress, Apoptosis and Cytoprotective Autophagy in Chicken Hepatocytes

**DOI:** 10.3390/toxins12020090

**Published:** 2020-01-29

**Authors:** Huadong Yin, Shunshun Han, Yuqi Chen, Yan Wang, Diyan Li, Qing Zhu

**Affiliations:** Farm Animal Genetic Resources Exploration and Innovation Key Laboratory of Sichuan Province, Sichuan Agricultural University, Chengdu 611130, Sichuan, China; yinhuadong@sicau.edu.cn (H.Y.); hanshunshun@stu.sicau.edu.cn (S.H.); chenyuqi@stu.sicau.edu.cn (Y.C.); as519723614@163.com (Y.W.); diyanli@sicau.edu.cn (D.L.)

**Keywords:** T-2 toxin, hepatocyte, apoptosis, autophagy, chicken

## Abstract

T-2 toxin is type A trichothecenes mycotoxin, which produced by fusarium species in cereal grains. T-2 toxin has been shown to induce a series of toxic effects on the health of human and animal, such as immunosuppression and carcinogenesis. Previous study has proven that T-2 toxin caused hepatotoxicity in chicken, but the regulatory mechanism is unclear. In the present study, we assessed the toxicological effect of T-2 toxin on apoptosis and autophagy in hepatocytes. The total of 120 1-day-old healthy broilers were allocated randomly into four groups and reared for 21 day with complete feed containing 0 mg/kg, 0.5 mg/kg, 1 mg/kg or 2 mg/kg T-2 toxin, respectively. The results showed that the apoptosis rate and pathological changes degree hepatocytes were aggravated with the increase of T-2 toxin. At the molecular mechanism level, T-2 toxin induced mitochondria-mediated apoptosis by producing reactive oxygen species, promoting cytochrome c translocation between the mitochondria and cytoplasm, and thus promoting apoptosomes formation. Meanwhile, the expression of the autophagy-related protein, ATG5, ATG7 and Beclin-1, and the LC3-II/LC3-I ratio were increased, while p62 was downregulated, suggesting T-2 toxin caused autophagy in hepatocytes. Further experiments demonstrated that the PI3K/AKT/mTOR signal may be participated in autophagy induced by T-2 toxin in chicken hepatocytes. These data suggest a possible underlying molecular mechanism for T-2 toxin that induces apoptosis and autophagy in chicken hepatocytes

## 1. Introduction

Mycotoxins are the main secondary metabolites of molds and lead to widespread contamination on crop plants and fruits. Among the most important mycotoxins, T-2 toxin is a mycotoxin that can cause multiple effects in organisms [1]. T-2 toxin is a type A trichothecene produced by several *Fusarium* species [2], which shows the most potent cytotoxicity [3]. Furthermore, T-2 toxin leads to the effects of cytotoxin radiomimetic, which is due to impaired protein synthesis. T-2 toxin hampers synthesis of DNA and RNA in eukaryotic cells, which ultimately triggers cell apoptosis in vitro and in vivo [4]. Many studies have shown that T-2 toxin induces apoptotic cell death in hematopoietic tissue [5], spleen, liver [6], skin and intestinal crypt in mice [7]. In chickens, apoptosis induced by T-2 toxin was detected in the thymus, bursa of Fabricius and primary hepatocytes [8,9]. Previous studies have demonstrated a crosstalk between autophagy and apoptosis, as apoptosis increases when the autophagic pathway is completely inhibited [10].

T-2 toxin contamination is usually found on cereals, such as maize, wheat and oats, which are the main food and feed resources for human and livestock [11]. The presence of T-2 toxin can be reduced but not completely eliminated. T-2 toxin can cause chronic toxicity in organisms after oral exposure, dermal exposure and inhalation. In livestock, this results in anorexia, reduced body weight and nutritional efficiency, altered neuro-endocrine system, and immune modulation [12]. In addition, residues of the T-2 toxin and its metabolites in animal products are an important human health problem. Poultry is extremely sensitive to the toxic effects of T-2 toxins, leading to yellow cheese-like necrosis at the edge of the septum, hard mucosal mucosa and typical angular cheilitis of the mouth and tongue [13]. In addition, chickens exposed to T-2 toxin show enhanced mortality from *Salmonella* infection and low-resistance titers for Newcastle disease and infectious bursal disease [14,15]. 

Multiple studies have examined the effects of T-2 toxin in inducing of hepatotoxicity in chickens. However, the relationship between T-2-induced autophagy and apoptosis has not been examined. Here, we investigated the effects of T-2 toxin on hepatocyte apoptosis and autophagy and provide experimental evidence for the potential molecular mechanism of T-2 toxin-induced hepatotoxicity in broiler chickens. 

## 2. Results 

### 2.1. Pathological Lesions

To determine the effect of T-2 toxin on chicken livers, we examined the pathomorphological changes in the liver. In the control group, the liver tissue structure was normal, the cell structure was intact, and the cells were arranged neatly (Figure 1A). In the 0.5 mg/kg T-2 toxin treatment group, the liver pathological changes were mild; the hepatocyte volume was increased and mild swelling manifested as blisters, with occasional inflammatory cell infiltration (Figure 1B). In the 1 mg/kg and 2 mg/kg treatment groups, the hepatocytes were swollen and showed balloon-like deformation; the cytoplasm was vacuolated, and the nucleus was located in the center of the vacuole or squeezed on one side. Additionally, hepatic sinus stenosis, a small amount of red blood cell deposits, focal inflammatory cell infiltration and massive proliferation of interlobular bile duct epithelial cells were observed in the 1 mg/kg and 2 mg/kg treatment groups (Figure 1C,D). 

### 2.2. T-2 Triggers Apoptosis in Hepatocytes

We next performed flow cytometry to determine if T-2 toxin induced apoptosis in hepatocytes from T-2 treated chickens. The amounts of apoptotic cells in the treatment groups were significantly higher (*p* < 0.01) than that in the control, and this difference was dose-dependent (Figure 2A,B). Western blot results showed cleavage of rapamycin (PARP) in the T-2 treatment groups; furthermore, pro-caspase-3 and pro-caspase-9 expressions were reduced in a dose-dependent manner, whereas the cleaved form of caspase-3 and caspase-9 increased (Figure 2C,D). These data further indicate that T-2 toxin induced apoptosis in hepatocytes.

### 2.3. The Mitochondrial Pathway is Activated by T-2 Toxin

To evaluate whether the mitochondrial pathway participates in the T-2 toxin-induced apoptosis, we first examined the mitochondrial reactive oxygen species (ROS) levels in hepatocytes from T-2 treated chickens by flow cytometry. Low intracellular ROS levels were found in the untreated group, whereas they increased dramatically in the 1 mg/kg and 2 mg/kg T-2 toxin treatment groups (Figure 3A,B). In addition, T-2 toxin significantly suppressed the activity of the antioxidant enzymes GSH-Px, CAT and SOD, but the MDA level was significantly higher in treatment groups than in the control group (Figure 3C). We next evaluated the protein expression of Bax and Bcl-2 and found that Bax protein abundance was upregulated, whereas Bcl-2 abundance was downregulated in a dose-dependent manner, with an increase in Bax/Bcl-2 ratio (Figure 3D). We also examined the mitochondrial release of cytochrome (cyt c) during T-2 toxin-induced apoptosis. The level of mitochondrial cyt c decreased with the increase of T-2 toxin concentration, whereas the level of cytosolic cyt c increased (Figure 3E).

### 2.4. T-2 Toxin Triggers Autophagy in Hepatocytes

To determine if T-2 toxin induces autophagy in hepatocytes from T-2 treated chickens, we measured the transcript levels of autophagy genes including ATG5, ATG7 and Beclin-1 genes (Figure 4A). T-2 toxin treatments induced greater expression levels of ATG5, ATG7 and Beclin-1 genes compared with controls. Furthermore, the ratio of LC3-II/LC3-I increased with the T-2 toxin dosage, while the protein abundance of p62 decreased (Figure 4B). In addition, the cell ultrastructure changed; typical autophagy features were observed and the number of autophagosomes increased in the treatment groups compared with controls (Figure 4C). 

### 2.5. Autophagy Protects Apoptosis in T-2 Treated Hepatocytes

Increased autophagy is considered a protective mechanism against apoptosis as both autophagy and apoptosis share common proteins. To explore the relationship between autophagy and apoptosis, the specific autophagy inhibitor 3-methyladenine (3MA) and autophagy inducer rapamycin (RAP) were used on T-2 toxin-treated hepatocytes. Immunofluorescence showed that T-2 toxin treatment significantly increased the numbers of LC3B puncta, and autophagy flux was further enhanced after the addition of RAP, but autophagy intensity was significantly decreased after the addition of 3MA (Figure 5A,B). When autophagy was inhibited by 3MA, the levels of caspase-3 and caspase-9 cleavage were significantly enhanced after T-2 treatment. Conversely, when autophagy was induced by RAP, the levels of caspase-3 and caspase-9 cleavage were significantly decreased (Figure 5C,D). These results may suggest that autophagy hinders apoptosis in T-2 toxin-treated hepatocytes.

### 2.6. T-2 Toxin Inhibits the PI3K/Akt/mTOR Signal Pathway

To determine if T-2 toxin regulates the PI3K/Akt/mTOR signal pathway in hepatocytes from T-2 treated chickens, we next examined the protein abundance of the tumor suppressor factor, phosphatase and tensin homolog (PTEN), which has a dual-specificity phosphatase activity. PTEN expression level increased in hepatocytes with the increase in T-2 toxin concentration (Figure 6A). In addition, we examined the protein abundance and phosphorylation levels of PI3K, Akt, mTOR and p70S6K, which are key proteins in the PI3K/Akt/mTOR pathway. We found that the protein abundances of PI3K, Akt, mTOR and p70S6K did not differ among the treatment groups, but their phosphorylation levels gradually decreased with the increase in T-2 toxin concentration (Figure 6B,C). These results may suggest that T-2 toxin inhibits the PI3K/Akt/mTOR signal pathway in hepatocytes.

## 3. Discussion 

The T-2 toxin has harmful mutagenic, carcinogenic and teratogenic effects on humans and animals [16,17,18]. Although various studies have examined hepatocyte apoptosis in broilers treated with T-2 toxin [19,20], no reports have focused on the relationship between autophagy and T-2 toxin-induced apoptosis. Herein, we reported that T-2 toxin-induced hepatotoxicity was characterized by the induction of mitochondrial-mediated apoptosis and PI3K/AKT/mTOR-mediated autophagy in chicken.

The liver is the main organ of metabolism in which foreign substances accumulate and are detoxified. The T-2 toxin suppresses hepatocyte protein synthesis and inhibits metabolic enzyme activity and liver fat peroxidation, which ultimately leads to hepatocyte apoptosis [21,22,23]. In the current study, histopathological analysis showed that T-2 toxin caused pathological changes in liver tissue, including hepatocyte swelling, volume increase and more granules in the cytoplasm, suggesting that T-2 toxin leads to hepatocyte apoptosis. Our results were consistent with the report by Meissonnier et al. who showed that exposure of pigs to T-2 toxin via diet for 28 days caused liver histopathological changes, excessive hepatic glycogen accumulation and mild interstitial inflammatory cell infiltration [24].

Apoptosis is a crucial physiological cell death process that can be induced by toxic stimuli [25]. Previous studies have shown that T-2 toxin injection can strongly induce cell apoptosis in different tissues, such as thymus, spleen and liver, particularly in the liver [6]. Yang et al. incubated primary chicken hepatocytes with T-2 toxin for 24 h and found that the cell activity was significantly reduced and apoptosis gradually increased in a dose-dependent manner [9], which was similar to our finding that hepatocyte apoptosis gradually increased with the increasing dosage of T-2 toxin.

The mitochondrial pathway has a vital role in the intrinsic apoptosis pathway [26], which depends on the translocation of the apoptogenic protein, cyt c, into the cytoplasm. This occurs via the Bax/Bcl-2 pathway, as their relative levels determine cell destiny by activating death-driving proteolytic proteins known as caspases [27]. In the current study, several findings suggested that T-2 toxin induced the mitochondrial apoptotic pathway in hepatocytes: (1) Bcl-2 was downregulated and Bax was upregulated, thus increasing the Bax/Bcl-2 ratio; and (2) cyt c was released from the mitochondria into the cytosol, followed by apoptosome formation with the apoptotic proteases Apaf-1 and caspase-9. In addition, T-2 toxin treatment lead to an increase in ROS and MDA levels and a decrease in the activities of SOD, CAT, and GSH-Px, resulting in oxidative stress and a concentration-dependent increase in apoptotic cells. Mu et al found that T-2 toxin can induce the ROS accumulation and an increase in mitochondrial mass, which indicated that oxidative stress and mitochondrial enhancement occurred in T-2 toxin-treated primary hepatocytes, which is similar to our result [28]. In addition to our results, other studies have shown apoptosis induced by T-2 toxin via the ROS-mediated mitochondrial pathway in other cells, such as ovarian granulosa cells [29], embryonic stem cells and fibroblast 3T3 cells [30] in mouse. 

Autophagy is a crucial homeostasis mechanism that is involved in multiple physiological and pathological processes [31]. Autophagy also shows a complex relationship with apoptosis, as autophagy not only increases caspase-dependent cell death, but also promotes cell survival [32]. In the present study, the increase in gene expression of Atg5, Atg7 and Beclin-1, which are autophagy marker genes, suggested that T-2 toxin induced autophagy in hepatocytes. Moreover, we found an increase in the LC3-II/LC3-I ratio and Beclin-1 protein abundance and a decrease in expression of p62 protein, further suggesting that T-2 toxin induced autophagy in hepatocytes. Bcl-2 and Beclin-1 participate in the regulation of both apoptosis and autophagy [33], and Bcl-2 interacts with Beclin-1 to suppress Beclin-1-dependent autophagy [34]. In our study, we found that Beclin-1 was activated by T-2 toxin, but Bcl-2 was suppressed, and T-2 toxin-induced apoptosis can be delayed by autophagy. Wang et al showed that autophagy may reduce zearalenone-induced cytotoxicity and prevent rat Leydig cell apoptosis [35]. Wu et al found that autophagy plays a role in protecting human cells from T-2 toxin-induced apoptosis, because autophagy may decrease toxic responses induced by T-2 toxin [36]. Our results were consistent with these reports.

The PI3K/AKT/mTOR/p70S6K signaling pathway plays a vital role in autophagy regulation in eukaryotic cells [37]. PI3K induces a signaling cascade and phosphorylates the serine/threonine kinase, mTOR, by activating the serine/threonine kinase, Akt [38]. PTEN has also been proven to suppress the Akt/mTOR signal [39]. As the major upstream modulator, the PI3K pathway regulates autophagy by phosphorylating AKT, which affects the downstream factors p70S6K and 4E-BP1 [40]. Several mycotoxins induce autophagy by inhibiting the PI3K/Akt/mTOR axis, such as zearalenone in donkey granulosa cells [41], aflatoxin B2 in chicken hepatocytes [16] and sterigmatocystin in human gastric epithelium cells [42]. In this study, T-2 toxin inhibited the phosphorylation of PI3K, Akt, mTOR and p70S6K, whereas it activated PTEN, suggesting that the PI3K/AKT/mTOR/p70S6K pathway may be participated in the autophagic process induced by toxicity effect of T-2 toxin. These findings are similar to a previous study that showed that deoxypodophyllotoxin induced cytoprotective autophagy against apoptosis through inhibition of the PI3K/AKT/mTOR pathway in osteosarcoma U2OS cells [42].

In summary, T-2 toxin treatment activates the mitochondrial apoptotic pathway by triggering ROS production and Bcl-2 family protein expression, resulting in hepatocyte apoptosis. In addition, T-2 toxin may involve in the PI3K/AKT/mTOR signal to regulate hepatocellular autophagy. This study provides new insights into the mechanisms underlying the toxicological effect of T-2 toxin in chicken hepatocytes. 

## 4. Materials and Methods

### 4.1. Ethics Approval

All experimental operations were approved by the Animal Ethics Committee of Sichuan Agricultural University, and the approved number was 2018-2121 (21 May 2018). Relevant guidelines and regulations were followed while performing all the methods.

### 4.2. Animals

A total of 120 ROSS 308 male chickens at one-day of age were used in this study. After being weighted, chickens were randomly divided into four groups (n = 30 per group); each treatment had six replicates with five chickens. Experimental replicates were raised in separate cages. The four groups were maintained under the same condition and received general nutrient composition and levels that met the requirement of ROSS 308, with T-2 toxin in feed as follows: 0 mg/kg (control), 0.5 mg/kg, 1 mg/kg, and 2 mg/kg. Feed and water were freely available during the whole trial period. 

### 4.3. Exposure of Chickens

All the feed was made up by the processing-workshop of feedstuff in the Animal Nutrition Institute of Sichuan Agricultural University, which meet the nutritional requirement of ROSS 308. There were no common mycotoxins, such as aflatoxins, deoxynivalenol, ochratoxin A, zearalenone and T-2 toxin, were found in this feed by the ELISA kit (Huaan Mangech Biotech, Beijing, China). Firstly, the T-2 toxin (purity ≥ 98%; Sigma Aldrich, St. Louis, MO, USA) powder was dissolved by 95% ethanol, and mixed in 1 kg feed and dry it. Then, the mixture was added into feed to get the get the target concentration (0 mg/kg, 0.5 mg/kg, 1 mg/kg, and 2 mg/kg, respectively) of T-2 toxin. At last, we used the ELISA kit (Huaan Mangech Biotech) to confirm the final concentration of toxins in the feed.

### 4.4. Sample Collection and Preparation

After 21 days of feeding, six chickens (one chicken for every replicate) were randomly selected from the same treatment and euthanized. Livers were collected to determine the pathological histology and hepatocyte apoptosis rate. Fresh livers were dissected, minced, and stored at − 80 °C for extracting RNA and protein. 

### 4.5. Pathological Observation

Liver tissues were fixed overnight in 4% phosphate-buffered paraformaldehyde (Jianke Biotech, Chengdu, Sichuan, China) and then paraffin-embedded blocks were archived. We sliced 5 μm thick tissue sections from paraffin-embedded tumor blocks and mounted the sections onto glass slides. Hematoxylin and eosin (H&E) staining was performed on tissue sections, and pathological examination was performed using an optical microscope (Olympus, Tokyo, Japan).

### 4.6. Apoptosis Detection

Livers were minced in pre-cold phosphate-buffered saline (PBS; Beyotime, Shanghai, China), and the suspension was passed through a 300 mesh nylon filter. After filtration, the hepatocyte suspensions were washed in PBS twice. Hepatocytes were re-suspended in 1× binding buffer (BD Pharmingen, Santiago, CA, USA) to obtain a concentration of 1 × 10^6^ cells/mL. Next, 100 μL were transferred into a culture tube and 5 μL of propidium iodide (PI; BD Pharmingen, Shanghai China) and 5 μL of Annexin V-FITC (BD Pharmingen, Shanghai, China) were added. After mixing, the cells were incubated at 25 °C for 15 min in the dark and then 400 μL of 1× binding buffer (BD Pharmingen, Shanghai, China) was added. Cells were then analyzed by FACSCanto II flow cytometry (BD Bioscience, San Diego, CA, USA).

### 4.7. Real-Time PCR

Total RNA of the livers were isolated by Trizol reagent (TaKaRa, Dalian, China). First-strand complementary cDNA was synthesized by PrimeScirptTM RT reagent kit with gDNA eraser (TaKaRa, Dalian, China) following the manufacturer’s protocol, and then was stored at − 20 °C for RT-PCR. PCR amplifications were performed as follows: 95 °C for 5 min and 36 cycles each with 95 °C for 10 s, 60 °C for 30 s and 72 °C for 20 s, then 65 °C for 5 s and 95 °C for 5 s using the BIO-RAD CFX Connect^TM^ real time system (Bio-Rad, Hercules, CA, USA). All PCR reactions were performed in triplicate. β-actin was used as the endogenous reference gene. Specific primers are referenced to Chen et al [16] or designed by the software of Primer Premier 5.0 (Ottawa, Ontario, Canada, 2007), and the primer sequences are listed in Table 1.

### 4.8. Western Blot Analysis

The refrigerated livers were washed with pre-cold PBS twice and centrifugation at 3000× *g* for 5 min at 4 °C, then removed the supernatant. Total protein extracts were obtained by homogenizing liver in RIPA lysis buffer (Sigma Aldrich) supplemented with protease inhibitor cocktail and phosphatase inhibitors. After centrifugation, the supernatant was collected and stored at − 80 °C. Protein concentration was determined by the BCA protein detection kit (Sangon Biotech, Shanghai, China). Western blot analysis was performed as previously described by Han et al. The primary antibodies were used: caspase-3 (ZenBio, Chengdu, China), caspase-9 (ZenBio), β-actin (Abcam, Cambridge, MA, USA), Bax (ZenBio), Bcl-2 (Santa Cruz, Heidelberg, Germany), LC3B (Sigma), P62 (Santa Cruz), beclin-1 (Sigma), PI3K/Akt/mTOR/70S6K protein and phosphorylated antibody were purchased from Bioss Biotechnology Co. Ltd. (Bioss, Beijing, China). The secondary antibodies used were as follows: mouse anti-rabbit (Sigma), goat anti-rabbit (Sigma), mouse anti-rabbit horseradish peroxidase (HRP) (Zenbio). The enhanced chemiluminescence (ECL) kit (Beyotime, Jiangsu, China) was used to capture the bands via a CanoScan LiDE 100 scanner (Canon, Tokyo, Japan), and western blots were analyzed by Image J software (Bethesda, MD, USA, 2007).

### 4.9. Cytochrome C Release

The cytoplasm was first isolated from the mitochondria using the cytochrome C release apoptosis kit (BioVision, Mountain View, CA, USA). After treatment with E2 for 24 h, the cells were lysed by homogenizing in the cytosol extraction solution provided by the kit and then centrifuged at 700× *g* for 10 min. Cells were then centrifuged at 12,000× *g* for 30 min to separate cytoplasmic and mitochondrial components. Determination of cytoplasmic and mitochondrial cytochrome C abundance was performed by western blot using mouse monoclonal antibodies provided in the kit.

### 4.10. Transmission Electron Microscopy (TEM) Observations

Hepatocytes were fixed in 2.5% glutaraldehyde phosphate buffer saline (Sigma, St. Louis, MO, USA) and post-fixed in 1% osmium tetroxide (Sigma). The samples were dehydrated in graded ethanol solutions, and cells were embedded in the stimulating resin. Sections (60 nm) were cut using ultramicrobody (Leica Microsystems, Milan, Italy). The divided grid has a saturated solution of uranyl acetate and lead citric acid. Samples were examined by electron microscopy (FEI, Milan, Italy).

### 4.11. Intracellular Reactive Oxygen Species (ROS) Detection 

Production of intracellular ROS production was measured using the fluorescent dye substrate 2’,7’-dichlorofluorescin-diacetate (DCFH-DA; Procell, Wuhan, China) as a substrate. Cells were incubated for 60 min at 37 °C with 10 μΜ DCFH-DA and then harvested and suspended in Hank’s Balanced Salt Solution (D-HBSS; Procell). The generation of ROS was analyzed using FACSCanto II flow cytometry (BD Bioscience, New York, NJ, USA).

### 4.12. Antioxidative Enzymes and Malondialdehyde Detection

The activities of superoxide dismutase (SOD), glutathione peroxidase (GPX-Px) and catalase (CAT) and malondialdehyde (MDA) level were determined by commercial assay kits (Jiancheng, Nanjing, China) according to the manufacturer’s instructions. After mixing the liver cell homogenate with the reagents, the cells were incubated at 37 °C overnight for multi-scan spectroscopy detection. 

### 4.13. Immunofluorescence and Confocal Microscopy

Hepatocytes grown on 24-well plates were fixed with 4% paraformaldehyde (Jianke Biotech, Guangzhou, China) for 10 min. After washing with PBS twice, cells were blocked using 3% bovine serum albumin (BSA; Thermo Fischer Scientific; former Savant, MA, USA) and 0.2% Triton X-100 (Thermo Fischer Scientific, Waltham, MA, USA) in PBS for 10 min at 37 °C. The samples were incubated with the relevant antibodies in PBS/10% FSC for 1 h and then stained with the appropriate fluorescent secondary antibody. Fluorescence intensities were captured by an Olympus FluoView FV1000 confocal microscope (Olympus, Melville, NY, USA). To block or induce autophagy, cells were treated with 3-methyladenine (10 mM; Sigma, St. Louis, MO, USA) or rapamycin (4 μM; Sigma), respectively, for 6 h. 

### 4.14. Statistical Analysis

Statistical analyses were performed using SPSS 19.0 software (SPSS Inc., Chicago, IL, USA, 2000). Data are shown as least squares means ± standard error of the mean (SEM). Differences between groups were assessed using t-test, and values were considered significant difference at *p* < 0.05.

## Figures and Tables

**Figure 1 toxins-12-00090-f001:**
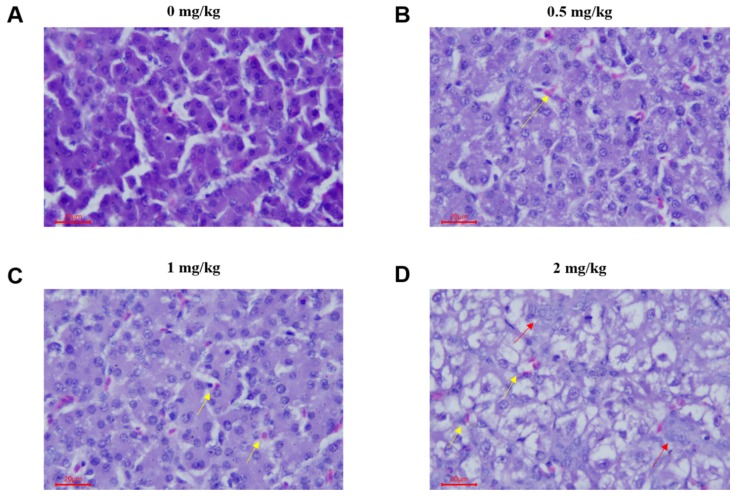
Photomicrographs of hematoxylin and eosin stained chicken liver sections of 21 day chicken after treatment of T-2 toxin with different concentration of 0, 0.5, 1 and 2 mg/kg. (**A**) No obvious pathological changes were observed in hepatocytes. (**B**) Hepatocytes with mild steatosis and slight congestion. (**C**) Hepatocytes were slightly swollen, with vacuolar degeneration and lymphocyte neutrophil infiltration. (**D**) The liver showed slight congestion, local vacuolar degeneration was obvious, and the bile duct epithelium and cells demonstrated slight hyperplasia. Red arrow: red blood cell; yellow arrow: bile duct epithelial cell; hematoxylin and eosin (H&E); bar, 20 μm.

**Figure 2 toxins-12-00090-f002:**
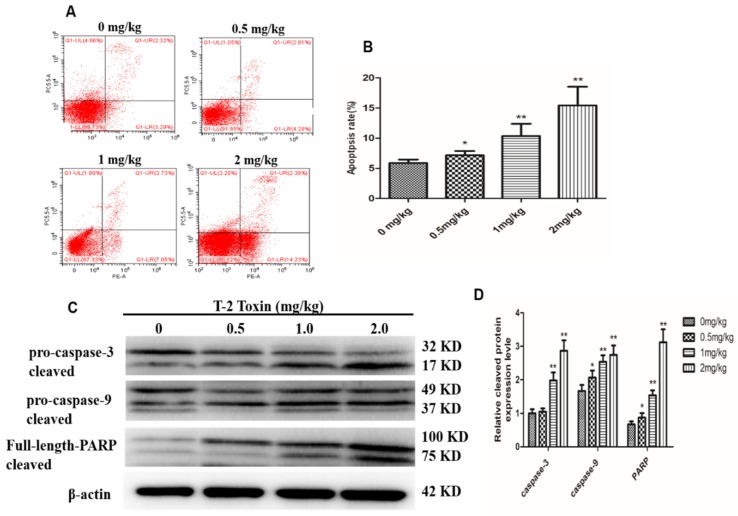
Effect of different concentration (0, 0.5, 1 and 2 mg/kg, respectively) of T-2 toxin on hepatocyte apoptosis. (**A**) Scattergram and (**B**) apoptosis rate of apoptotic hepatocytes. (**C**) The protein levels of PARP, caspase-3 and caspase-9, and their cleaved forms in hepatocytes. (**D**) The bar showed the relative protein cleaved level of caspase-3, caspase-9 and PARP. The data are presented as the means ± standard error of the mean (SEM) of three independent experiments. * *p* < 0.05 and ** *p* < 0.01, compared with the control group.

**Figure 3 toxins-12-00090-f003:**
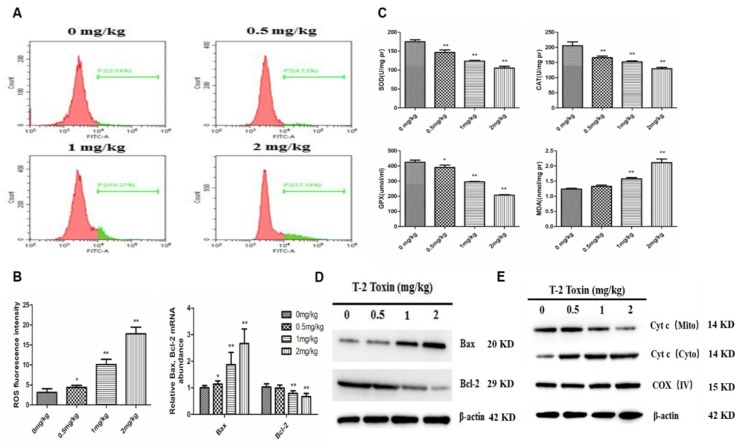
T-2 toxin induced hepatocyte apoptosis via activation of the mitochondria-dependent pathway. (**A,B**) Intracellular reactive oxygen species (ROS) levels in hepatocytes from chickens treated with T-2 toxin of different concentration at 0, 0.5, 1 and 2 mg/kg. (**C**) The activity of antioxidant enzymes SOD, CAT, GPX-Sh and MDA content in hepatocytes from chickens treated with T-2 toxin of different concentration at 0, 0.5, 1 and 2 mg/kg. (**D**) The Bax and Bcl-2 mRNA and protein levels in hepatocytes from chickens treated with T-2 toxin of different concentration at 0, 0.5, 1 and 2 mg/kg. (**E**) The cytosolic and mitochondrial cyt c level in hepatocytes from chickens treated with T-2 toxin of different concentration at 0, 0.5, 1 and 2 mg/kg. All the data are presented as means ± SEM of three independent experiments. * *p* < 0.05 and ** *p* < 0.01, compared with the control group.

**Figure 4 toxins-12-00090-f004:**
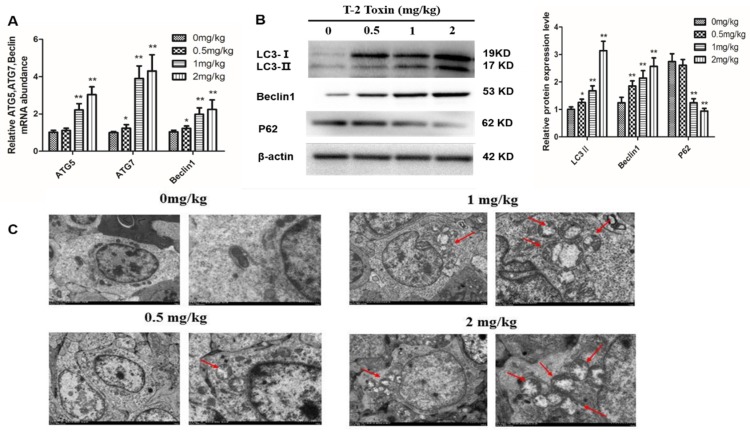
Effect of T-2 toxin on autophagy in chicken hepatocytes. (**A**) The mRNA levels of Beclin-1, Atg5 and Atg7 in hepatocytes from chickens treated with T-2 toxin of different concentration at 0, 0.5, 1 and 2 mg/kg. (**B**) The protein expression levels of LC3, p62 and Beclin-1 in hepatocytes from chickens treated with T-2 toxin of different concentration at 0, 0.5, 1 and 2 mg/kg. (**C**) Morphological observation of autophagy in hepatocytes from chickens treated with T-2 toxin of different concentration at 0, 0.5, 1 and 2 mg/kg, autophagic vacuoles (red arrows, magnification from left to right: ×1200, ×5000). All the data are presented as means ± SEM of three independent experiments. * *p* < 0.05 and ** *p* < 0.01, compared with the control group.

**Figure 5 toxins-12-00090-f005:**
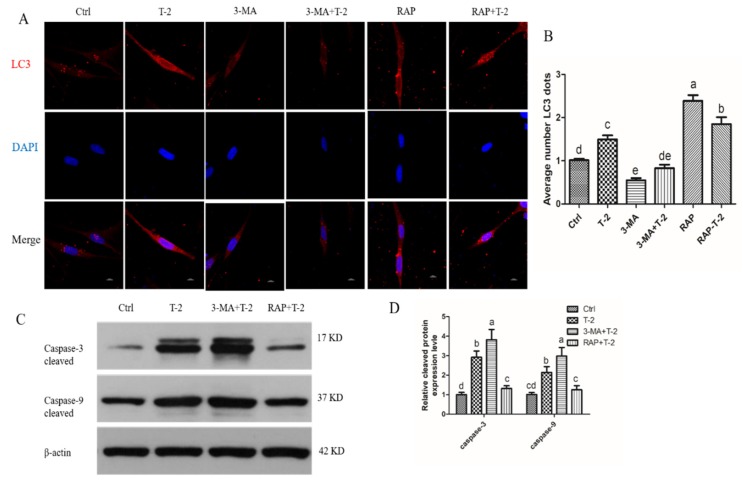
Autophagy delays apoptosis in T-2 treated hepatocytes. (**A**) Hepatocytes stained with LC3 (red) antibody using a confocal microscope (600x), Nuclei were stained with 4,6-diamino-2-phenyl indole (DAPI) (blue; bar = 10 μm). (**B**) The bar showed the number of LC3 dots. (**C**) Western blots showed the expression levels of caspase-3 and caspse-9 cleaved in hepatocytes. (**D**) The bar showed the protein level of cleaved caspase-3 and caspase-9. All the data are presented as means ± SEM of three independent experiments. * *p* < 0.05 and ** *p* < 0.01, compared with the control group.

**Figure 6 toxins-12-00090-f006:**
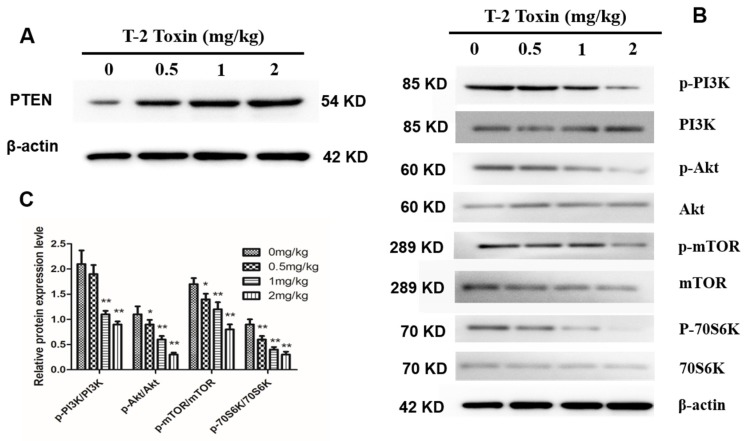
Effect of T-2 toxin on PI3K/Akt/mTOR in hepatocytes. (**A**) Protein abundance of phosphatase and tensin homolog (PTEN) in hepatocytes from chickens treated with T-2 toxin of different concentration at 0, 0.5, 1 and 2 mg/kg. (**B**) Representative blots showed the expression abundance of p-PI3K, PI3K, p-Akt, Akt, p-mTOR, mTOR, p-70S6K and 70S6K in hepatocytes from chickens treated with T-2 toxin of different concentration at 0, 0.5, 1 and 2 mg/kg. (**C**) The bar graphs showed the ratio of p-PI3K/PI3K, p-Akt/Akt, p-mTOR/mTOR, and p-p70S6K/p70S6K. All the data are presented as means ± SEM of three independent experiments. * *p* < 0.05 and ** *p* < 0.01, compared with the control group.

**Table 1 toxins-12-00090-t001:** Table: Gene-special primers for RT-PCR.

Gene	Forward Primer	Reverse Primer	NCBI Accession no	Tm/℃	Product/bp
Bcl-2	5-ATCGTCGCCTTCTTCGAGTT-3	5-ATCCCATCCTCCGTTGTCCT-3	NM_205339.2	61	78
Atg5	5-GATGAAATAACTGAAAGGGAAGC-3	5-TGAAGATCAAAGAGCAAACCAA-3	NM_001006409.1	52	124
Atg7	5-TCAGATTCAAGCACTTCAGA-3	5-GAGGAGATACAACCACAGAG-3	NM_001030592.1	55	62
Beclin-1	5-CAGACACGCTGCTGGACC-3	5-TCTCCTTGTCATCCTCGTTCA-3	NM_001006332.1	60	84
β-actin	5-CCGCTCTATGAAGGCTACGC-3	5-CTCTCGGCTGTGGTGGTGAA-3	NM_204313.1	60	127

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
