# Peer review of "T-2 Toxin Induces Oxidative Stress, Apoptosis and Cytoprotective Autophagy in Chicken Hepatocytes"

_toxins, 2020, doi:10.3390/toxins12020090_

Round 1

Reviewer 1 Report

Dear Authors,

I found your manuscript relevant and interesting. However, the English is dramatic and need to be corrected by a native speaker. I also have some doubts about the Figure captions- should providing all information necessary to understand without coming back to the main text as well as Discussion section. The manuscript does not cover the template of Toxins and should be corrected. Other comments you might find in PDF file.

Best regards,

Reviewer 2 Report

Dear Authors,

the article titled "T-2 Toxin Induces Mitochondrial-mediated Apoptosis and PI3K/AKT/mTOR-mediated Autophagy in Chicken Hepatocytes" provided a lot of useful information about influence of T-2 toxin on mitochondrial-mediated apoptosis and PI3K/AKT/mTOR-mediated autophagy in chicken hepatocytes. The article is well-written and deserve publication after a few improvements.

Point 1: As far as I can see the primers for Bax gene mentioned in Table 1 were previously used in the paper Chen et al. 2016, please attach the citation for Bax primers.

Point 2: In section 2.8. please include that liver tissue was homogenized before.

Point 3: Please specify which kit was used in sections 2.9 and 2.12.

Point 4: I would like to see the photos of the whole membranes used for western blot analysis with molecular ladder on it. Additionally on the figures 2.c, 3.d.e, 4.b, 5.c and 6.a please include the molecular weight of the bands.

Reviewer 3 Report

The paper entitled: T-2 Toxin Induces Mitochondrial-mediated Apoptosis and PI3K/AKT/mTOR-mediated Autophagy in Chicken Hepatocytes, present new data concerning the apoptosis and the autophagy induced by T2 toxin in chicken hepatocytes. The paper is very complex and shows many interesting results. However in the results sections there are some experiments that are not described in the Material & Methods section and some section of the paper are unclear and confuse. The English language need extensive revision.

Specific comments

Introduction/aim of the paper

What is the relevance of the used doses of the toxin in term of occurrence of T2 toxin in poultry feed?

Material and methods –

2.3 Exposure of Chichens. The presence of other mycotoxins than T2 toxin has been verified in food? Please add som details/explanations

2.7. Real time PCR Table 1. Please insert Gene accessions number, Tm and amplicon length

2.8 This section of the M&M is confuse as in the results section beside the western blot results for caspase 3 and 9 there are many other results of western blots (PARP,.Bax, Bcl2, LC3I/II, p62). For these, the experimental design was not presented in M&M section. Also more details about the calculation and interpretation of the results obtained for cleaved pro-caspase 3 and 9

It is not clear when the liver lysates were used for the western blor analyse and if isolated hepatocytes or cell cultures were also used.

Line 137 2.13 How were the hepatocytes grown on the coverslips. Please provide details

Line 140. What means relevant antibodies.. please provide their names and provenience.

Figure 5. To which experiment belong the analyses of caspases?

Figure 6. The same as for the figure 5.

Minor comments

Abstract, line 5. Please replace fusarium with Fusarium

Introduction, line 47 Please replace salmonella with Salmonella and low resistance titers with low resistance antibody titers

Material and Methods

T2 toxins was provided from Sigma (line 64-65) or Sicchuan Yuanou Biotechnology (line 69)

Line 68 2.3 Exposure of Chichens to be replaced with Exposure of chickens to mycotoxin

Line 77 Please replace the pathological histology with the hstopathological changes

Line 77. What mean some fresh liver? The liver samples were not collected from all the euthanized animals and were not used for all the investigations?

Line 83. What means the first tissue section? The microscope was used to examine the preparation and not the - pathological examination was used by the microscope!

In all figure legends, the +/- sign is missing.

Figure 4. Please add in the fig 4 legend the meaning of the red arrow

Reviewer 4 Report

General comments

The aim of the study „T-2 Toxin Induces Mitochondrial-mediated Apoptosis and PI3K/AKT/mTOR-mediated Autophagy in Chicken Hepatocytes” is to evaluate the effects and potential mechanism of action of T-2 toxin on chicken hepatocytes. The authors have observed that T-2 toxin induces intrinsic apoptotic pathway in hepatocytes through inhibition of Bcl-2 and upregulation of Bax and release of cyt c from the mitochondria into the cytosol, followed by apoptosome formation. Furthermore, exposure to T-2 toxin leads to increase in reactive oxygen species and malonil dialdehyde levels, and to decrease in the activities of antioxidant enzymes. The authors have also observed that T-2 toxin increases the expression of autophagy marker  genes Atg5, Atg7 and Beclin-1 and increase in the LC3-II/LC3-I ratio and the Beclin-1 protein level,  decrease expression of p62 protein thus promoting  autophagy in  hepatocytes. Further, they showed that autophagy interfere with apoptosis in T-2 treated hepatocytes. Moreover, the authors showed inhibition of the PI3K/Akt/mTOR signaling pathway.

The results are very interesting. The manuscript is well-written and its organization is well structured.

Specific comments

(1)   Author names are missing. 

(2)   Lines 196,197, 225 and 242: Please specify whether the hepatocytes or animals have been treated with T-2 toxin?

(3)   Lines 216-217: It remained unclear whether the authors were treating hepatocytes or animals with specific autophagic inhibitor 3-methyladenine and inducer rapamycin? How long was the treatment? What doses were used?  

(4)   Line 231: 3.6. T-2 toxin induces autophagy in hepatocytes through inhibition of the PI3K/Akt/mTOR signal: How did the authors conclude that autophagy was mediated through PI3K/Akt/mTOR signaling? Additional experiments are needed to confirm this statement.

(5)   Published results on the effect of T-2 toxins on human liver cells (L02) (Wu J, Zhou Y, Yuan Z, et al. Autophagy and Apoptosis Interact to Modulate T-2 Toxin-Induced Toxicity in Liver Cells. Toxins (Basel). 2019;11(1):45. doi:10.3390/toxins11010045) should be taken into consideration while discussing the findings of the present experiments.

Round 2

Reviewer 1 Report

Dear Authors,

I appreciate the effort made to revise this publication.

Best reards,

Reviewer 3 Report

Dear Authors

I consider that the quality of the manuscript was significantly improved after the last corrections. The manuscript can be published in its present form in Toxins journal.

Best regards,

Reviewer 4 Report

I sought additional experiments because the authors reached the conclusion without experiments (the authors conclude that autophagy was mediated through PI3K /Akt /mTOR signaling without experimental support). The authors did not perform additional experiments. However, they tone down the statement in Result section of the manuscript. Such approach is also acceptable. Still, from my point of view, in order to be acceptable for publication, the authors also need to tone down the statement in the title of manuscript, abstract (line 19) and discussion (line 335). This needs to be corrected.

Also, for the legends of Figure 3 (A, B and C); Figure 4 (A, B and C) and Figure 6 (A, B and C) the authors were responded that the hepatocytes were extracted form the livers that collected from the chickens treated with T-2 toxin. However, the authors did not corrected this in the manuscript. Please correct this.